# Homeostatic microglia initially seed and activated microglia later reshape amyloid plaques in Alzheimer's Disease

Nóra Baligács [1,2], Giulia Albertini [1,2], Sarah C. Borrie [1,2], Lutgarde Serneels[1,2], Clare Pridans [3,4], Sriram Balusu[1,2] & Bart De Strooper [1,2,5] ✉

The role of microglia in the amyloid cascade of Alzheimer's disease (AD) is debated due to conflicting findings. Using a genetic and a pharmacological approach we demonstrate that depletion of microglia before amyloid-β (Aβ) plaque deposition, leads to a reduction in plaque numbers and neuritic dystrophy, confirming their role in plaque initiation. Transplanting human microglia restores Aβ plaque formation. While microglia depletion reduces insoluble Aβ levels, soluble Aβ concentrations stay consistent, challenging the view that microglia clear Aβ. In later stages, microglial depletion decreases plaque compaction and increases neuritic dystrophy, suggesting a protective role. Human microglia with the *TREM2^{R47H/R47H}* mutation exacerbate plaque pathology, emphasizing the importance of non-reactive microglia in the initiation of the amyloid cascade. Adaptive immune depletion (*Rag2^{-/-}*) does not affect microglia's impact on plaque formation. These findings clarify conflicting reports, identifying microglia as key drivers of amyloid pathology, and raise questions about optimal therapeutic strategies for AD.

Alzheimer's disease (AD) is a neurodegenerative disorder characterized by several pathological features including the aggregation of amyloid-beta (Aβ) peptides in both brain parenchyma and vasculature, aggregation of hyper-phosphorylated TAU protein within neurons, neuroinflammation, synaptic dysfunction, and progressive neuronal loss. The classical model of AD progression suggests that the disease begins with the accumulation of misfolded proteins, predominantly the amyloid-beta peptide (Aβ), forming extracellular plaques[1]. This phase, also referred to as the biochemical phase[2], involves the build-up of amyloid plaques leading to complex responses by astrocytes and microglia, neuronal pathology, and network changes, called the cellular phase of AD[2]. Finally, after decades of silent pathology, AD enters the clinical phase wherein cognitive and functional symptoms manifest, typically around twenty years after amyloid plaques can first be detected. The amyloid cascade hypothesis is supported by genetic

evidence that autosomal dominant mutations causing early-onset AD affect the amyloid precursor protein or its processing enzymes, causing early plaque accumulation[3]. Recent progress with passive immunotherapy demonstrated clinical benefits by reducing amyloid plaques from the brains of patients[4,5]. However, the vast majority of AD cases are not caused by autosomal dominant mutations. Still, the genetic contribution to sporadic AD risk is estimated to be about 70%[6]. This genetic risk for late-onset AD is largely associated with microglia, indicating a crucial role of these cells in modulating the disease[7,8].

Genome-wide association studies have identified over 80 risk variants associated with AD[6]. These variants are predominantly or highly expressed in microglia[7,9,10]. However, the precise role of microglia in AD pathogenesis remains elusive. On one hand, microglia are believed to facilitate amyloid clearance through phagocytosis, potentially limiting plaque pathology[11–13]. On the other hand, microglia may also drive

[1]Centre for Brain and Disease Research, Flanders Institute for Biotechnology (VIB), Leuven, Belgium. [2]Department of Neurosciences and Leuven Brain Institute, KU Leuven, Leuven, Belgium. [3]University of Edinburgh Centre for Inflammation Research, Edinburgh, UK. [4]Simons Initiative for the Developing Brain, University of Edinburgh, Edinburgh, UK. [5]UK Dementia Research Institute at UCL, University College London, London, UK. ✉e-mail: b.destrooper@ukdri.ucl.ac.uk

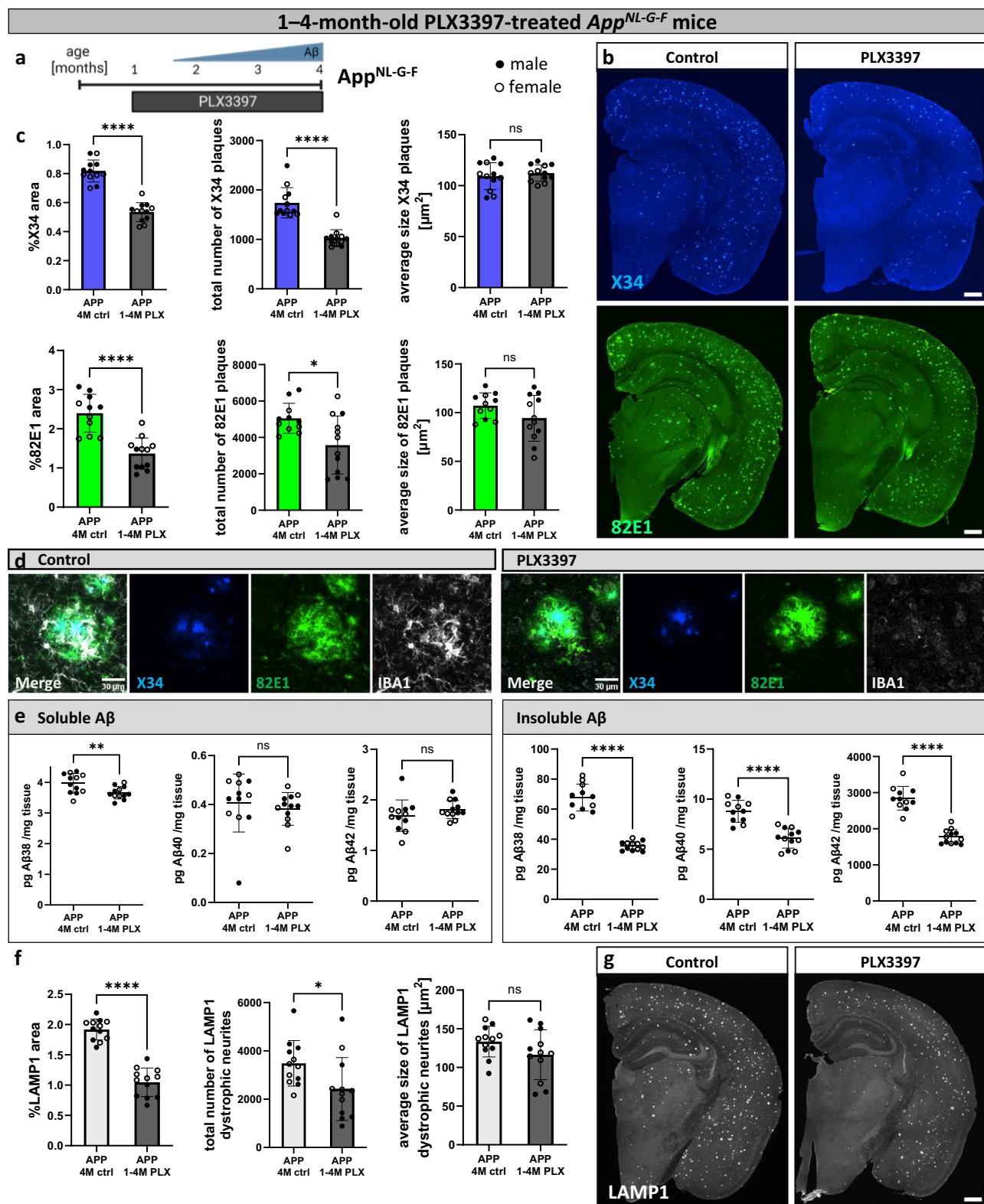

chronic neuroinflammation, exacerbating neuronal damage[14] and even promote amyloid plaque formation and spreading[12,15,16]. In response to amyloid plaque accumulation, microglia cluster around amyloid plaques, adapt an activated morphology and a distinct transcriptional signature, referred to as disease-associated microglia (DAM), microglial neurodegenerative phenotype (MGnD), or activated response microglia (ARM)[17–19]. The response in human microglia encompasses in addition cytokine (CRM) and HLA responses[20]. The genetic risk of AD,

assessed by the expression of associated risk genes, is spread over all different cell states of microglia, including homeostatic microglia[20,21]. Despite the well-documented microglial cellular states, their molecular functions remain unclear.

Studies investigating microglial depletion through genetic ($Csf1r^{\Delta FIRE/\Delta FIRE}$)[22] or pharmacological inhibition of the colony-stimulating factor 1 receptor (CSF1R) signaling[23] have yielded conflicting results regarding their role in amyloid plaque pathology. While

**Fig. 1 | Microglia seed early amyloid plaques. a** Treatment scheme. $App^{NL-G-F}$ mice (APP) were fed PLX3397 from 1 month until analysis at 4 months of age (1−4 M PLX) or a control diet (4 M ctrl). Created in BioRender. Cherretté, E. (2024) https://BioRender.com/u17o338. **b** Representative confocal images of amyloid plaques in the brain, stained with X34 (fibrillar plaques) or 82E1 (total Aβ). **c** Quantification of amyloid plaques in the whole brain (%X34 area: two-tailed t-test, $n$(ctrl) = 12, $n$(PLX) = 12, $p$ = 0.0000000016; total number of X34 plaques: two-tailed Mann-Whitney test, $n$(ctrl) = 12, $n$(PLX) = 12, $p$ = 0.000002; average size X34 plaques: two-tailed t-test, $n$(ctrl) = 12, $n$(PLX) = 12, $p$ = 0.5098; %82E1 area: two-tailed d t-test, $n$(ctrl) = 11, $n$(PLX) = 12, $p$ = 0.000015; total number of 82E1 plaques: two-tailed t-test with Welch's correction, $n$(ctrl) = 11, $n$(PLX) = 12, $p$ = 0.012; average size 82E1 plaques: two-tailed t-test, $n$(ctrl) = 11, $n$(PLX) = 12, $p$ = 0.1225). **d** Higher magnification images of amyloid plaques and microglia in control and PLX3397-treated mice. **e** ELISA of amyloid levels in soluble and insoluble cortex extracts (sol. Aβ38: two-

tailed t-test, $n$(ctrl) = 12, $n$(PLX) = 12, $p$ = 0.0098; sol. Aβ40: two-tailed Mann-Whitney test, $n$(ctrl) = 12, $n$(PLX) = 12, $p$ = 0.1725; sol. Aβ42: two-tailed t-test, $n$(ctrl) = 12, $n$(PLX) = 12, $p$ = 0.2571; insol. Aβ38: two-tailed t-test with Welch's correction, $n$(ctrl) = 11, $n$(PLX) = 12, $p$ = 0.00000009; insol. Aβ40: two-tailed t-test, $n$(ctrl) = 11, $n$(PLX) = 12, $p$ = 0.000004; insol. Aβ42: two-tailed t-test, $n$(ctrl) = 11, $n$(PLX) = 12, $p$ = 0.000000005). **f** Quantifications of dystrophic neurites around amyloid plaques (%LAMP1 area: two-tailed t-test, $n$(ctrl) = 12, $n$(PLX) = 12, $p$ = 0.0000000007; total number of LAMP1 dystrophic neurites: two-tailed t-test, $n$(ctrl) = 12, $n$(PLX) = 12, $p$ = 0.0317; average size of LAMP1 dystrophic neurites: two-tailed t-test, $n$(ctrl) = 12, $n$(PLX) = 12, $p$ = 0.1321). **g** Representative images of LAMP1⁺ dystrophic neurites in the brain. White dots represent female mice and black dots represent male mice. Scale bars 500 μm (**b**, **g**), 30 μm (**d**). All data is presented as mean ± SD. *$p$ ≤ 0.05; **$p$ ≤ 0.01; ***$p$ ≤ 0.001; ****$p$ ≤ 0.0001. Source data are provided as a Source Data file.

some studies suggest that microglia depletion can inhibit plaque formation[24,25], others report no change[26,27] or even an increase in plaque size[11,28] (Supplementary Table 1). These conflicting studies underscore the need for further research to clarify whether microglia are beneficial or detrimental to AD pathology[29] and whether they act primarily during the early biochemical phase of amyloid deposition or play a more prominent role in the later cellular phase, potentially modulating the progression of AD pathology or contributing to synaptic and neuronal damage[2].

In addition to microglia, the adaptive immune system has also been implicated in AD progression[30]. Dysregulation of T cells has been observed in the blood and CSF of AD patients[31,32]. CD8⁺ T cells were found in proximity to Aβ plaques and neuronal processes[33]. However, research on the interactions between adaptive immunity and microglia in the context of amyloid pathology is still limited. Notably, studies using *Rag2⁻/⁻, which* results in an impaired adaptive immune system in mice[34] have produced conflicting results with increased amyloid plaque pathology in *Rag2⁻/⁻*; 5xFAD mice[35], and reduced pathology and decreased brain Aβ levels in *Rag2⁻/⁻*; PSAPP mice[36]. Hence, the role of adaptive immunity in amyloid plaque formation in AD remains controversial.

Given these complexities, we hypothesized that microglia may perform distinct functions at different stages of amyloidosis, influenced by their dynamic cellular states. To test this, we employed PLX3397[23] mediated microglial depletion both before and after amyloid plaque deposition in $App^{NL-G-F}$ mice. Additionally, we explored the role of peripheral adaptive immunity in the microglia-amyloid interaction by depleting microglia in $App^{NL-G-F}$; $Rag2^{-/-}$ mice. To validate our findings, we utilized a genetic model of microglia depletion ($Csf1r^{ΔFIRE/ΔFIRE}$)[22] and investigated the effects of human microglia transplantation, including those carrying the $TREM2^{R47H/R47H}$ AD risk variant, which fail to respond to amyloid plaque pathology. Our findings highlight the complex disease stage- and cell-state-dependent functions of microglia in AD. Microglia are critical in the initiation of AD, specifically in the formation of amyloid plaques, challenging the prevailing view that microglia primarily serve to clear Aβ. In more advanced stages, activated microglia compact amyloid plaques and limit their toxic effects on neurons.

## Results

### Microglia seed early amyloid plaques
To understand the role of microglia in the early stages of amyloid formation, we depleted microglia in $App^{NL-G-F}$ mice by targeting the CSF1R using a pharmacological antagonist. $App^{NL-G-F}$ mice were treated with PLX3397 in chow from 1 month (before plaque pathology) until 4 months of age (Fig. 1a). The amyloid burden and associated neuritic dystrophy were evaluated. Microglia depletion was greater than 80%, as evaluated by IBA1 staining (Supplementary Fig. 1a). We used two readouts for amyloid plaque formation at 4 months of age, X34 is a dye that reacts with β-sheeted amyloids, and 82E1 is a monoclonal antibody against the Aβ peptide recognizing all conformations of

Aβ plaques. Both X34-positive amyloid fibrils and 82E1-immunoreactive Aβ deposits were reduced in the PLX3397-treated group. Microglia depletion reduced the area covered by amyloid plaques and the number of plaques but not their average size (Fig. 1b−d). ELISA measurements of soluble and insoluble Aβ brain fractions from the cortex confirmed this conclusion: soluble Aβ levels remained stable (apart from a minor decrease in Aβ38), while significant reductions of Aβ38, Aβ40, and Aβ42 were observed in the insoluble fraction (Fig. 1e). The reduction in amyloid plaques was accompanied by a reduction in LAMP1⁺ dystrophic neurites (Fig. 1f, g). Thus microglia are seeding new plaques during the early amyloidosis phase in AD and cause increased neuritic dystrophy. The observation that soluble Aβ levels remain stable when microglia are depleted, suggests that Aβ turnover and clearance is largely independent of microglia.

### Microglia compact late amyloid plaques
To investigate the role of microglia in more advanced stages of amyloidosis, $App^{NL-G-F}$ mice were treated with PLX3397 after plaques had been generated, beginning at 3 months until 7 months of age (Fig. 2a). In line with previous findings that activated microglia are less dependent on CSF1R signaling[19,37], microglia depletion was less effective, however, IBA1 staining area was reduced by more than 60% (Supplementary Fig. 1b). 82E1-positive amyloid plaques increased significantly in area and size, but not in number (Fig. 2b−d). The levels of soluble and insoluble Aβ in the brain were unchanged, except for a reduction in insoluble Aβ38 (Fig. 2e). Notably, insoluble Aβ42 levels in the cortex of $App^{NL-G-F}$ mice are 10-fold higher than insoluble Aβ38 levels, hence the reduction in Aβ38 may be negligible. Similar to the 82E1-positive amyloid plaque staining, the LAMP1-staining for dystrophic neurites was increased in area and size (Fig. 2f, g). 82E1 and LAMP1 staining areas across the early and late treatment cohorts and controls were highly correlated (Supplementary Fig. 1c). This indicates that the decrease in dystrophic neurite pathology is likely not a direct effect of the microglia on the neurons, but the indirect consequence of plaque compaction, decreasing the exposure of the neurons to amyloid plaques. In conclusion, microglia do not alter the overall Aβ load in the brain but compact amyloid plaques and reduce dystrophic neurites in the late stage of amyloid plaque accumulation.

Sustained microglia depletion from before the onset of amyloid plaques (1 month) until 7 months of age (Supplementary Fig. 1d−h), resulted in a combination of the early seeding and late compacting effects of microglia on amyloid plaques. The total number of plaques was reduced due to reduced seeding, while the average plaque size was increased due to reduced compaction. Overall, this treatment regimen resulted in reduced amyloid area and insoluble Aβ levels in the brain (Supplementary Fig. 1e−g).

### The adaptive immune system does not alter microglia-mediated modulation of amyloid plaque pathology
To test the role of adaptive immunity in modulating amyloid plaques and microglia-plaque interactions, we depleted microglia with

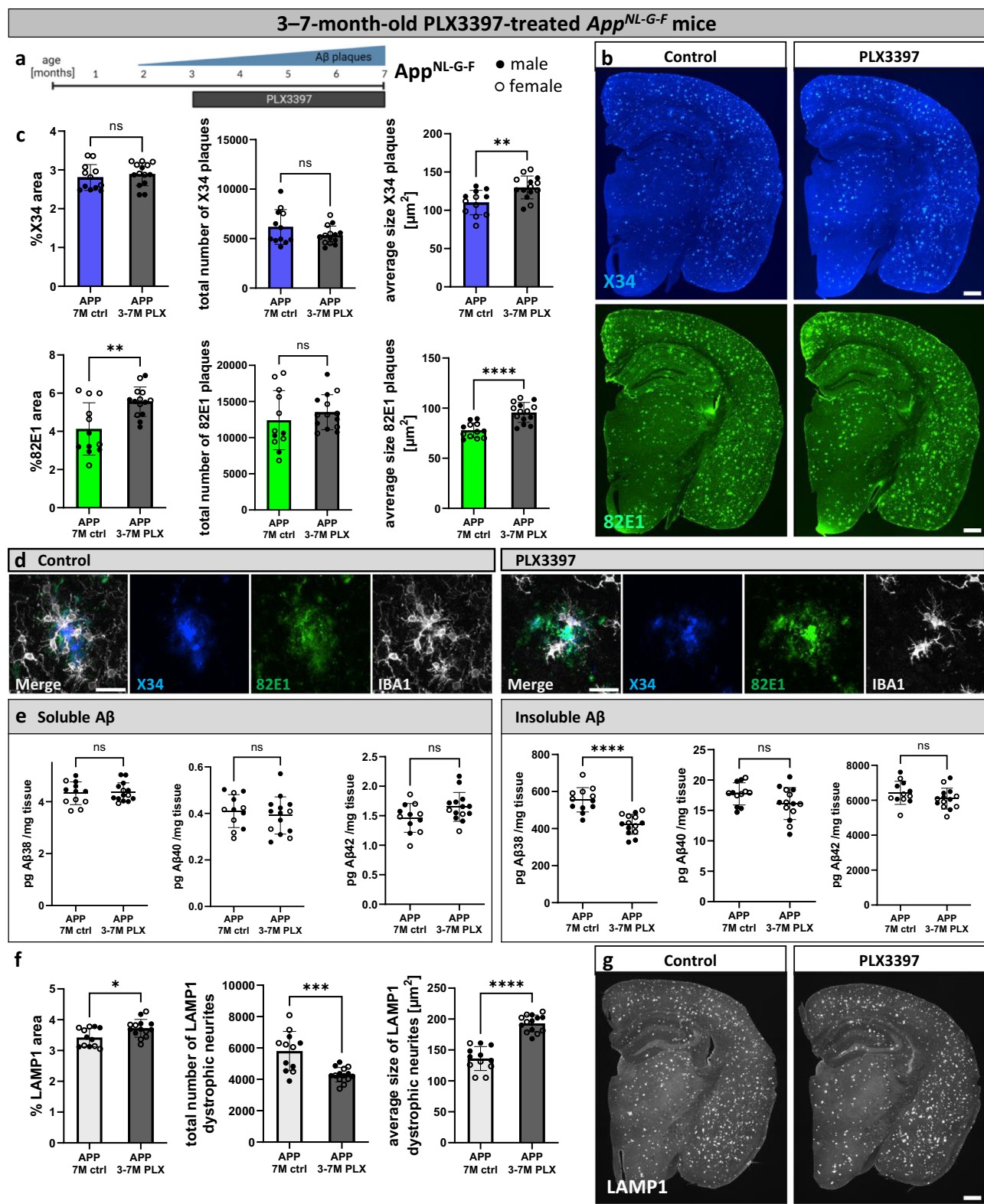

PLX3397 in immunodeficient $App^{NL-G-F}$; $Rag2^{-/-}$ mice. As in the immune-competent mice (Fig. 1), we found a reduction of amyloid plaque size, number, and insoluble Aβ levels when microglia were depleted early in amyloid plaque deposition (1–4 M) in immunodeficient $App^{NL-G-F}$; $Rag2^{-/-}$ mice compared to $App^{NL-G-F}$; $Rag2^{-/-}$ mice with intact microglia (Fig. 3a−c). When microglia were depleted at more advanced stages of amyloidosis in immunodeficient mice (3−7 M), amyloid plaque area and size were increased (Fig. 3d−f), again

matching the effects in immunocompetent mice (Fig. 2). Soluble and insoluble Aβ levels were unchanged except for a decrease in insoluble Aβ38. Sustained microglia depletion also resulted in similar effects as observed in $App^{NL-G-F}$ mice. Plaque numbers were reduced, while the average plaque size increased, resulting in a net decrease in plaque area and insoluble Aβ levels (Supplementary Fig. 2c−g). Finally, the overall soluble and insoluble Aβ concentrations were similar in $App^{NL-G-F}$; $Rag2^{-/-}$ mice compared to $App^{NL-G-F}$ mice across all

**Fig. 2 | Microglia compact late amyloid plaques. a** Treatment scheme. *App^{NL-G-F}* mice (APP) were fed PLX3397 from 3 months until analysis at 7 months of age (3−7 M PLX) or control diet (7 M ctrl). Created in BioRender. Cherretté, E. (2024) https://BioRender.com/u17o338. **b** Amyloid plaques in the brain, stained with X34 (fibrillar plaques) and 82E1 (total Aβ). **c** Image quantifications of amyloid plaques in the whole brain (%X34 area: two-tailed t-test, n(ctrl) = 12, n(PLX) = 14, p = 0.4926; total number of X34 plaques: two-tailed t-test with Welch's correction, n(ctrl) = 12, n(PLX) = 14, p = 0.1575; average size X34 plaques: two-tailed t-test, n(ctrl) = 12, n(PLX) = 14, p = 0.0034; %82E1 area: two-tailed t-test, n(ctrl) = 12, n(PLX) = 14, p = 0.0027; total number of 82E1 plaques: two-tailed t-test, n(ctrl) = 12, n(PLX) = 14, p = 0.4042; average size 82E1 plaques: two-tailed t-test, n(ctrl) = 12, n(PLX) = 14, p = 0.00003). **d** Higher magnification images of amyloid plaques and microglia in control and PLX3397-treated mice. **e** ELISA of amyloid levels in soluble and insoluble cortex extracts (sol. Aβ38: two-tailed t-test, n(ctrl) = 12, n(PLX) = 14, p = 0.7964; sol. Aβ40: two-tailed t-test, n(ctrl) = 12, n(PLX) = 14, p = 0.5302; sol. Aβ42: two-tailed t-test, n(ctrl) = 12, n(PLX) = 14, p = 0.0638; insol. Aβ38: two-tailed t-test, n(ctrl) = 12, n(PLX) = 14, p = 0.000008; insol. Aβ40: two-tailed t-test, n(ctrl) = 12, n(PLX) = 14, p = 0.0887; insol. Aβ42: two-tailed t-test, n(ctrl) = 12, n(PLX) = 14, p = 0.1904). **f** Quantifications of dystrophic neurites around amyloid plaques (% LAMP1 area: two-tailed t-test, n(ctrl) = 12, n(PLX) = 14, p = 0.0172; total number of LAMP1 dystrophic neurites: two-tailed t-test, n(ctrl) = 12, n(PLX) = 14, p = 0.0003; average size of LAMP1 dystrophic neurites: two-tailed t-test, n(ctrl) = 12, n(PLX) = 14, p = 0.000000006). **g** LAMP1⁺ dystrophic neurites in the brain. White dots represent female mice and black dots represent male mice. Scale bars 500 μm (**b**, **g**), 30 μm (**d**). All data is presented as mean ± SD. *p ≤ 0.05; **p ≤ 0.01; ***p ≤ 0.001; ****p ≤ 0.0001. Source data are provided as a Source Data file.

treatments. This indicates that adaptive immunity does not alter the effect of microglia on amyloid plaque load or total amyloid levels in the *App^{NL-G-F}* model.

### FIRE mice fail to initiate amyloid plaques which are restored by transplantation of human microglia

Microglia depletion with pharmacological inhibitors is incomplete. To exclude effects by PLX3397-resistant microglia or off-target effects of the drug, we sought an independent way to confirm the effects of microglia depletion on amyloid plaque pathology. We generated a deletion of the fms-intronic regulatory element (FIRE) enhancer in the Csf1r gene (*Csf1r^{ΔFIRE/ΔFIRE}*) of *App^{NL-G-F}; Rag2^{−/−}; IL2rg^{−/−}; hCSF1KI* mice. These mice develop amyloid plaques, allow for the transplantation of human microglia[20,38,39], and completely lack microglia (Fig. 4a)[22], hereafter referred to as FIRE mice. We xenotransplanted human-derived microglial progenitors[20,39] into these mice at P4, and collected the brains 3 or 6 months after transplantation. Human transplanted microglia populate the entire mouse brain (Supplementary Fig. 3a). Additionally, staining with the DAM-marker CD9 is sparse and of low intensity in human microglia in 3-month-old FIRE mice (Supplementary Fig. 3b). By the age of 6 months, however CD9-staining is widespread and much more intense, where human CD9⁺ microglia cluster around amyloid plaques (Supplementary Fig. 3c). This indicates that human microglia display a delayed response to amyloid plaque deposition.

*App^{NL-G-F}; Rag2^{−/−}; IL2rg^{−/−}; hCSF1^{KI}* mice (control, with intact mouse microglia) show amyloid plaque deposition already at 6 weeks of age as demonstrated by 82E1 and X34 staining, which is not seen in FIRE mice (Fig. 4a). This result was confirmed by a 77% reduction in insoluble Aβ42 in the absence of microglia at the same age (Fig. 4b). Apolipoprotein E (ApoE), a major genetic risk factor for AD, has been shown to play an important role in amyloid plaque formation[40]. ApoE is upregulated once microglia become activated by exposure to amyloid plaques[17,19,41]. However, in his early stage, microglia are not activated yet, which was confirmed by ApoE staining (Supplementary Fig. 4) showing that at this stage of pathology, ApoE is mainly expressed by astrocytes[41,42]. By 3 months of age, FIRE mice develop some amyloid plaques, but to a much lower extent than control mice (Fig. 4c−e), while grafting of human microglia significantly restored plaque burden in the FIRE mice (Fig. 4c−e). Although the restoration is not complete, the result confirms the important contribution of microglia to amyloid plaque seeding at 3 months of age. By 6 months, the differences in amyloid load and dystrophic neurites between the control and FIRE mice groups are less pronounced but FIRE mice still have significantly reduced plaque load, insoluble Aβ levels (Supplementary Fig. 5a−c), and LAMP1⁺ dystrophic neurites (Supplementary Fig. 5d, e) compared to control mice. In contrast, levels of insoluble Aβ42 in xeno-transplanted FIRE mice have reached similar levels as in control mice (Supplementary Fig. 5c). In conclusion, these results independently confirm the crucial role of microglia in initiating amyloid plaques in early disease stages.

### Human *TREM2^{R47H/R47H}* microglia exacerbate amyloid and dystrophic neurite pathology

The experiments above demonstrate the dual interaction of microglia with amyloid plaque load. Our data suggest that microglia mediate the seeding of plaques even when they are in their homeostatic state, while compaction of amyloid plaques is performed by microglia that become activated after exposure to amyloid plaques. To test this further we grafted H9-derived WT human microglia and *TREM2^{R47H/R47H}* microglia[43] into the FIRE mice. The triggering receptor expressed on myeloid cells 2 (TREM2) is a transmembrane receptor abundantly expressed on microglia. Loss of function mutations and polymorphisms of the *TREM2* gene (e.g., TREM2 R47H) are strong risk factors for AD[10,44,45] and impair the microglia response to plaques[20,46−49].

At 6-7 weeks of age, both WT and *TREM2^{R47H/R47H}* xenotransplanted human microglia seeded amyloid plaques in FIRE mice (Supplementary Fig. 6), which were not present in non-grafted FIRE mice (Fig. 4a, b), confirming that non-reactive microglia are capable to seed amyloid plaques. The plaque load at that age was very low and not significantly different between WT and *TREM2^{R47H/R47H}* microglia but the observation nevertheless reinforces the importance of microglia for the initial seeding of the amyloid plaques. Interestingly, at 3 months, insoluble Aβ42 levels were significantly increased in *TREM2^{R47H/R47H}* microglia-grafted FIRE brains compared to WT microglia-grafted brains, suggesting increased seeding capacity of these microglia (Fig. 5c). Furthermore, the size of X34⁺ and 82E1⁺ plaques was increased as well, indicating that the late protective function in plaque compaction is also affected in the *TREM2^{R47H/R47H}* microglia (Fig. 5a, b). This was accompanied by an increase in the LAMP1⁺ dystrophic neurite area (Fig. 5d−f). Thus *TREM2^{R47H/R47H}* mutations, which prevent the activation of microglia in response to amyloid plaques, exacerbate amyloid pathology both by increasing the seeding of plaques in early amyloid deposition, and not being able to build the protective response once the plaques are formed.

## Discussion

The role of microglia in AD is multifaceted. In this study, we examined microglia's involvement in the amyloid cascade of AD, revealing a dual-phase response. We showed that microglia operate both upstream and downstream of amyloid plaque formation, assuming two distinct cell states: homeostatic microglia facilitate plaque formation, while activated microglia compact plaques in later stages of the disease and limit neuritic dystrophy.

Early microglia depletion, prior to plaque onset in the *App^{NL-G-F}* model, led to reduced levels of insoluble Aβ and fewer plaques without affecting plaque size. This supports the notion that microglia are key in initiating plaque formation. These results were validated using an independent model of microglia depletion (FIRE mice), where human microglia transplantation restored amyloid plaque seeding, demonstrating microglia's direct involvement in early plaque initiation. Once plaques were formed, microglia depletion did not alter Aβ levels or plaque numbers but did impact plaque morphology, indicating that

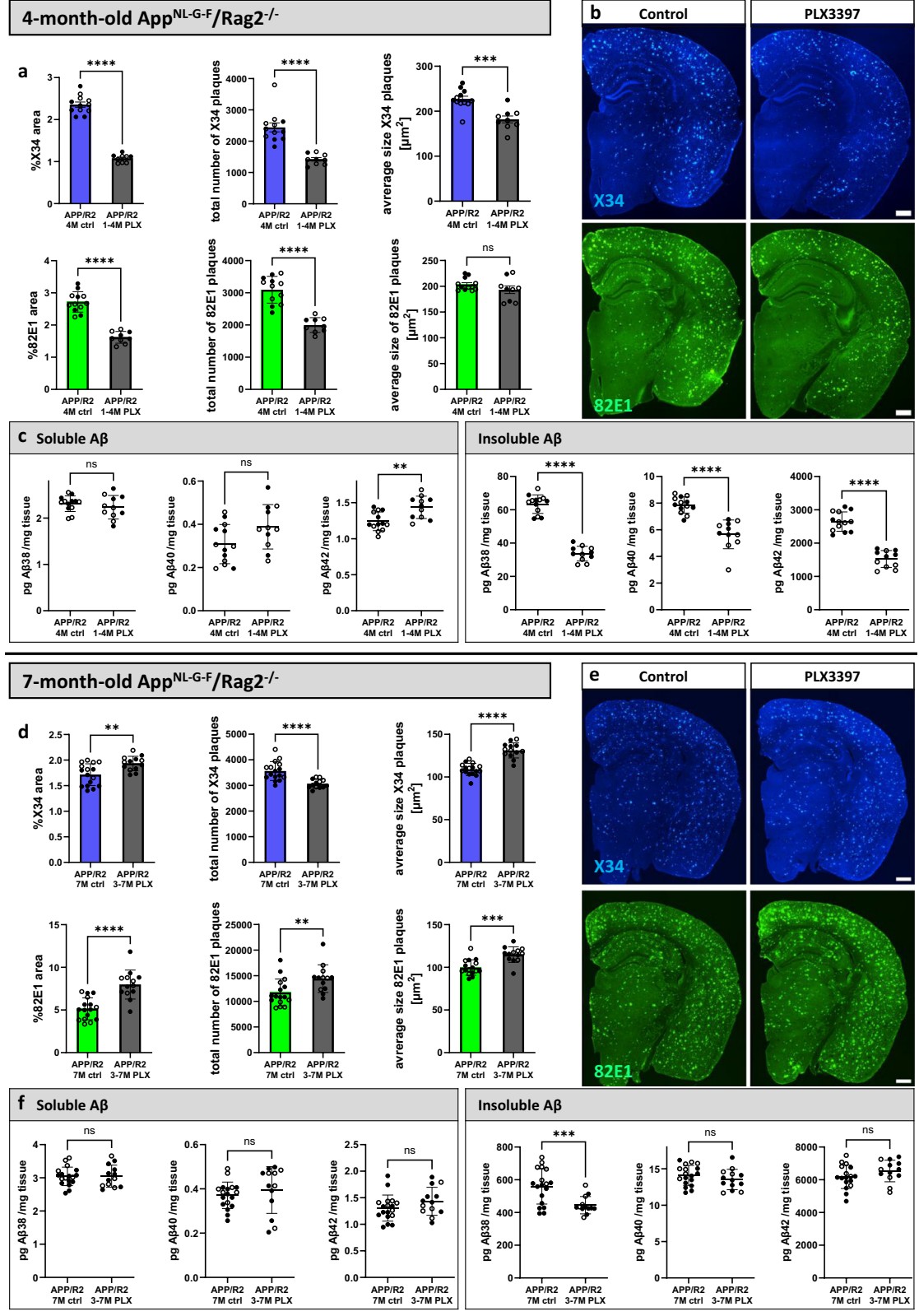

microglia also play a pivotal role in compacting plaques during advanced stages of disease. The close correlation between dystrophic neurite and amyloid staining suggests that the effects of microglia on neurites are indirect by modifying the amyloid plaque exposure to neurons. Plaque compaction by microglia appears therefore protective.

We further confirmed our findings through two pathology-relevant approaches. First, transplantation of human microglia with the *TREM2*[R47H/R47H] risk variant, known to prevent microglial activation[20,46–49], exacerbated amyloid plaque and neuritic pathology compared to WT human microglia. This indicates that microglia activation is beneficial both during the early plaque formation and later plaque compaction stages of AD, although neuroinflammation may still harm other aspects of the disease. Second, we explored whether microglial expression of ApoE, a key genetic risk factor for AD, is

**Fig. 3 | The adaptive immune system does not alter microglia-mediated modulation of amyloid plaques. a–c** Immunodeficient $App^{NL-G-F}/Rag2^{-/-}$ mice (APP/R2) were fed PLX3397 from 1 month until analysis at 4 months of age (1–4 M PLX) or control diet (4 M ctrl). **a** Quantifications of amyloid plaques in the whole brain (%X34 area: two-tailed t-test with Welch's correction, $n$(ctrl) = 12, $n$(PLX) = 9, $p$ = 0.000000000005; total number of X34 plaques: two-tailed Mann-Whitney test, $n$(ctrl) = 12, $n$(PLX) = 9, $p$ = 0.000007; average size X34 plaques: two-tailed t-test, $n$(ctrl) = 12, $n$(PLX) = 9, $p$ = 0.0002; %82E1 area: two-tailed t-test, $n$(ctrl) = 12, $n$(PLX) = 9, $p$ = 0.00000002; total number of 82E1 plaques: two-tailed t-test, $n$(ctrl) = 12, $n$(PLX) = 9, $p$ = 0.0000009; average size 82E1 plaques: two-tailed t-test, $n$(ctrl) = 12, $n$(PLX) = 9, $p$ = 0.0926). **b** Representative images of amyloid plaques stained with X34 and 82E1. **c** Aβ ELISA of soluble and insoluble cortex extracts (sol. Aβ38: two-tailed t-test, $n$(ctrl) = 13, $n$(PLX) = 11, $p$ = 0.3824; sol. Aβ40: two-tailed t-test, $n$(ctrl) = 13, $n$(PLX) = 11, $p$ = 0.0558; sol. Aβ42: two-tailed t-test, $n$(ctrl) = 13, $n$(PLX) = 11, $p$ = 0.0029; insol. Aβ38: two-tailed t-test, $n$(ctrl) = 13, $n$(PLX) = 11, $p$ = 0.000000000001; insol. Aβ40: two-tailed Mann-Whitney test, $n$(ctrl) = 13, $n$(PLX) = 11, $p$ = 0.000002; insol. Aβ42: two-tailed t-test, $n$(ctrl) = 13, $n$(PLX) = 11, $p$ = 0.000000001). **d–f** Immunodeficient $App^{NL-G-F}/Rag2^{-/-}$ mice (APP/R2) were treated with PLX3397 from 3 months until analysis at 7 months of age (3–7 M PLX) or control diet (7 M ctrl). **d** Image quantifications of amyloid plaques (%X34 area: two-tailed t-test, $n$(ctrl) = 16, $n$(PLX) = 13, $p$ = 0.0036; total number of X34 plaques: two-tailed t-test with Welch's correction, $n$(ctrl) = 16, $n$(PLX) = 13, $p$ = 0.00009; average size X34 plaques: two-tailed t-test, $n$(ctrl) = 16, $n$(PLX) = 13, $p$ = 0.00000004; %82E1 area: two-tailed t-test, $n$(ctrl) = 16, $n$(PLX) = 13, $p$ = 0.00002; total number of 82E1 plaques: two-tailed Mann-Whitney test, $n$(ctrl) = 16, $n$(PLX) = 13, $p$ = 0.0056; average size 82E1 plaques: two-tailed t-test, $n$(ctrl) = 16, $n$(PLX) = 13, $p$ = 0.0001. **e** Representative images of amyloid plaques stained with X34 and 82E1. **f** Aβ ELISA of soluble and insoluble cortex extracts (sol. Aβ38: two-tailed d t-test, $n$(ctrl) = 18, $n$(PLX) = 13, $p$ = 0.9646; sol. Aβ40: two-tailed t-test with Welch's correction, $n$(ctrl) = 18, $n$(PLX) = 13, $p$ = 0.4748; sol. Aβ42: two-tailed t-test, $n$(ctrl) = 18, $n$(PLX) = 13, $p$ = 0.1756; insol. Aβ38: two-tailed t-test with Welch's correction, $n$(ctrl) = 18, $n$(PLX) = 13, $p$ = 0.0006; insol. Aβ40: two-tailed t-test, $n$(ctrl) = 18, $n$(PLX) = 13, $p$ = 0.2682; insol. Aβ42: two-tailed t-test, $n$(ctrl) = 17, $n$(PLX) = 13, $p$ = 0.1715). White dots represent female and black dots represent male mice. Scale bars 500 μm (**b**, **e**). All data is presented as mean ± SD. $*p \leq 0.05$; $**p \leq 0.01$; $***p \leq 0.001$; $****p \leq 0.0001$. Source data are provided as a Source Data file.

involved in amyloid plaque formation[40]. Immunostaining in 6-week-old mice (Supplementary Fig. 4) confirmed previous findings that under homeostatic conditions, astrocytes are the primary providers of ApoE in the CNS[41,42]. ApoE expression in microglia is only detected later in the disease when they adopt an inflammatory state[17,19,41]. Our observations are in line with work by Henningfield et al. who showed that a microglia-specific ApoE deletion does not alter plaque formation[50]. However, microglia may take up APOE from their surrounding[51], which serves as a co-factor for plaque formation[52].

Studies before have implicated microglia in the formation of amyloid plaques[12,15,16], but the consensus in the field is that microglia clear Aβ from the brain[12,13]. Our study resolves many of the contradictions in the previous literature (Supplementary Table 1), where microglia were variably described as either promoting or mitigating amyloid pathology. By carefully considering disease stages and microglial cell states, we demonstrate that early-stage microglia initiate plaque formation, while later-stage microglia perform plaque compaction.

Importantly, the seeding and compaction of amyloid plaques by microglia are independent of the adaptive immune system, as similar effects were observed in immunodeficient $App^{NL-G-F}$; $Rag2^{-/-}$ mice. This indicates that while adaptive immunity may modulate other aspects of AD[30,35], microglia's role in amyloid plaque dynamics operates independently. This is in line with our previous RNA sequencing data analyzing the effect *of* $Rag2^{-/-}$ deficiency on transcriptomic profiles in the brain of amyloid plaque AD mice[53].

Interestingly, despite significant changes in insoluble Aβ levels, which reflect amyloid plaque accumulation, soluble Aβ levels remained relatively unchanged across different ages and experimental conditions. This indicates that Aβ turnover is only marginally dependent on microglia and that the amyloid plaque accumulation is driven by microglia presence rather than increasing soluble Aβ concentration.

Our discovery that homeostatic microglia initiate amyloid plaque seeding has critical implications, as it suggests that the genetics of sporadic AD may partially converge on amyloid initiation similarly to familial AD. Homeostatic microglia uptake extracellular Aβ and concentrate it within endosomes and lysosomes at an acidic pH[54,55], which could generate intracellular high molecular weight species[56,57]. In the interstitial and cerebrospinal fluid, the pH is neutral, and Aβ concentrations are comparatively low, which does not promote Aβ aggregation[58,59]. Studies have shown that microglia can uptake and release Aβ[60], produce amyloid fibrils[52,61] or propagate amyloid plaques[15,62]. These processes may be accelerated by AD risk genes, many of which are implicated in endocytosis, phagocytosis or the lysosomal system[7,8]. The release of seeds may be mediated by microglial cell death[15,57,62] but further studies are needed. This mirrors the behavior of macrophages in other amyloidogenic diseases[63,64], suggesting that the mechanism has broad biomedical relevance.

In summary, our data place microglia at the very origin of the amyloid cascade, and future studies should explore how AD risk genes expressed in homeostatic microglia[20], may prime these cells for early plaque formation. Moreover, the therapeutic potential of microglia activation[63,64] in promoting plaque clearance is supported by recent antibody trial results[65], suggesting that activation of microglia may counteract amyloid pathology. Previously it was thought that activated microglia were detrimental to AD[66,67].

In conclusion, we demonstrate that non-reactive microglia seed plaques early in AD, while activated microglia later compact plaques, limiting their toxicity. Our findings clarify previous contradictory reports on microglial function and highlight the need to consider disease stages and microglial states when developing AD therapies.

## Methods
### Mice
All rodent experiments were approved by the Ethical Committee for Animal Experimentation (ECD) of KU Leuven and were executed in compliance with the ethical regulations for animal research. $App^{NL-G-F}$ mice ($App^{tm3.1Tcs+}$; Takaomi Saido)[68] express amyloid precursor proteins (APP) at endogenous levels but contain the humanized Aβ sequence, as well as Swedish (NL; K670_M671delinsNL), Arctic (G; E693G), and Iberian (F; I716F) familial Alzheimer's disease-causing mutations in C57BL/6 background. This strain was crossed with homozygous $Rag2$ knockout mice ($Rag2^{tm1.1Cgn}$; Jackson Laboratory, strain 008309) to generate $App^{NL-G-F}/Rag2^{-/-}$ mice. Homozygous $Rag2^{tm1.1Flv}$; $Csf1^{tm1(CSF1)Flv}$; $Il2rg^{tm1.1Flv}$ mice (Jackson Laboratory, strain 017708) were crossed with $App^{tm3.1Tcs+}$ to generate the $Rag2^{tm1.1Flv}$; $Csf1^{tm1(CSF1)Flv}$; $Il2rg^{tm1.1Flv}$; $App^{tm3.1Tcs}$ strain. To generate the FIRE mice, homozygous mouse oocytes from $Rag2^{tm1.1Flv}$; $Csf1^{tm1(CSF1)Flv}$; $Il2rg^{tm1.1Flv}$; $App^{tm3.1Tcs}$ crosses[15] in a mixed C57Bl6; BALB/c; 129S4 background were microinjected with reagents targeting the fms-intronic regulatory sequence (FIRE sequence) in the intron 2 of the mouse Csf1R gene as previously described by David Hume and Clare Pridans[22]. Ribonucleoproteins containing 0.3 μM purified Cas9HiFi protein 0.3 μM crRNA (5′GTCCCTCAGTGTGTGAGA3′ and 5′CAATGAGTCTGTACTGGAGC3′) and 0.3 μM trans-activating crRNA (Integrated DNA Technologies) were injected into the pronucleus of 120 embryos by the CBD Mouse Expertize Unit of KU Leuven. One female founder with the expected 428 bp deletion was selected and crossed with a $Rag2^{tm1.1Flv}$; $Csf1^{tm1(CSF1)Flv}$; $Il2rg^{tm1.1Flv}$; $App^{tm3.1Tcs}$ male and the progeny were interbred to obtain a $Rag2^{tm1.1Flv}$; $Csf1^{tm1(CSF1)Flv}$; $Il2rg^{tm1.1Flv}$; $App^{tm3.1Tcs}$; $Csf1R^{em1Bdes}$. For maintaining the colony $Rag2^{-/-}$; $Csf1^{CSF1/CSF1}$; $Il2rg^{-/y}$; $App^{NL-G-F/NL-G-F}$; $Csf1R^{\Delta FIRE/\Delta FIRE}$ males were crossed with $Rag2^{-/-}$; $Csf1^{CSF1/CSF1}$; $Il2rg^{-/-}$; $App^{NL-G-F/NL-G-F}$; $Csf1R^{\Delta FIRE/WT}$ females as the 5 times

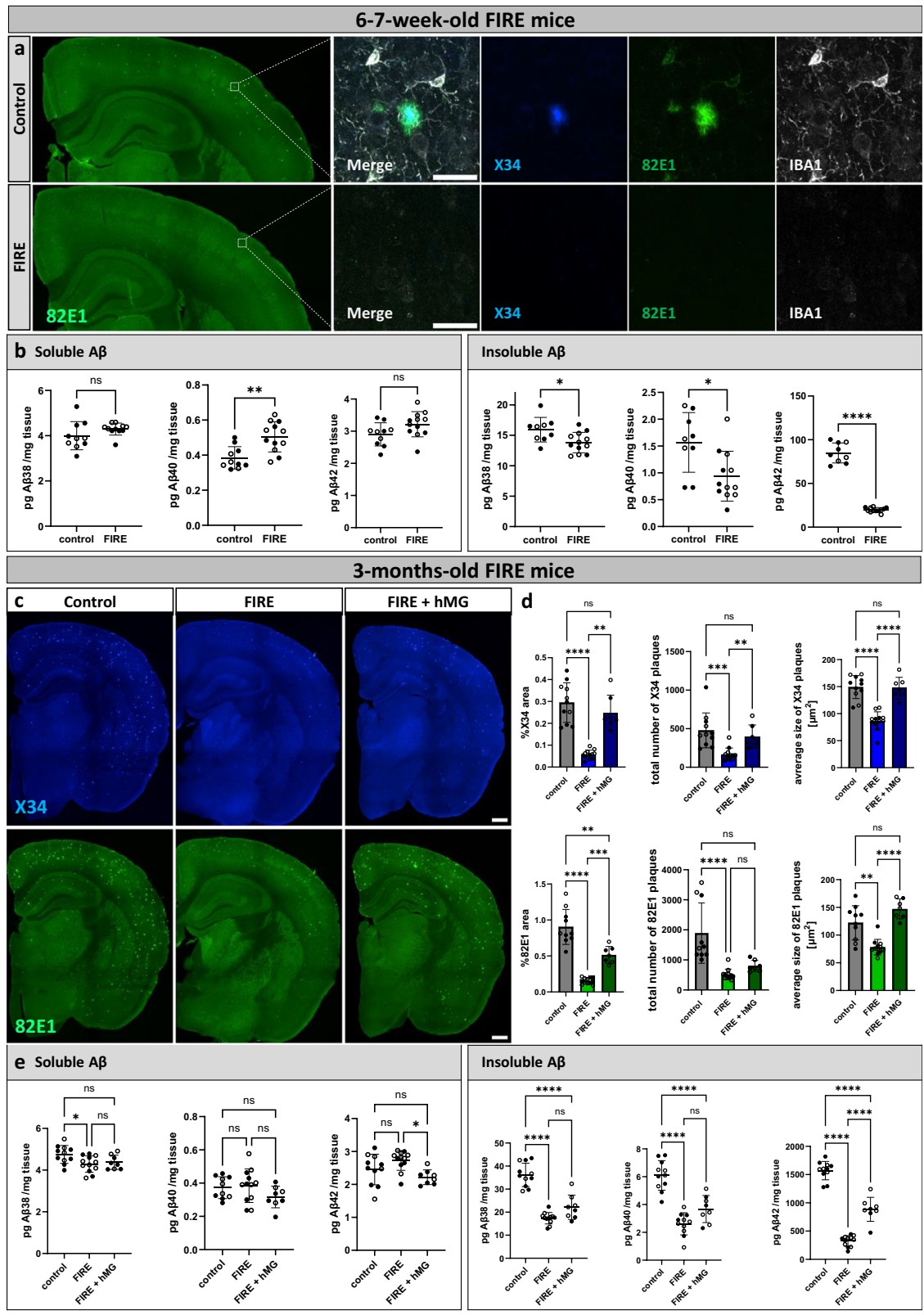

homozygous females tend to take less care of their progeny. Mice were randomized, and both sexes were used for all experiments. Mice were housed in groups of two to five per cage with ad libitum access to food and water and a 14 h light/10 h dark cycle at 21 °C and 32% humidity. PLX3397 (Pexidartinib PLX3397, Asclepia MedChem Solutions), a CSF1R antagonist, was mixed in the mouse chow (600 mg/kg), which was replaced two times per week during the treatment period.

## Immunofluorescence and Imaging
Mice were sacrificed with an overdose of sodium pentobarbital and immediately perfused with ice-cold 1 × DPBS (Gibco, Cat.#14190-144). After perfusion, one hemisphere was postfixed for 24 h at 4 °C in 4% formaldehyde solution (PFA) for histology. The second hemisphere was dissected on ice. The olfactory bulb and cerebellum were removed, and the hippocampus, cortex, and midbrain were

**Fig. 4 | Genetic depletion of microglia delays amyloid plaque deposition and is reversed by xenotransplantation of human microglia. a** Overview and higher magnification images of amyloid plaques in the brain and microglia in 6–7-week-old control mice ($App^{NL-G-F}$; $Rag2^{-/-}$; $IL2rg^{-/-}$; $hCSF1^{KI}$, with mouse microglia) and FIRE mice ($App^{NL-G-F}$; $Rag2^{-/-}$; $IL2rg^{-/-}$; $hCSF1^{KI}$; $Csf1r^{\Delta FIRE/\Delta FIRE}$, no microglia). **b** Aβ ELISA of soluble and insoluble cortex extracts in 6–7-week-old control and FIRE mice (sol. Aβ38: two-tailed t-test with Welch's correction, $n$(ctrl) = 10, $n$(FIRE) = 11, $p$ = 0.1936; sol. Aβ40: two-tailed t-test, $n$(ctrl) = 10, $n$(FIRE) = 12, $p$ = 0.0016; sol. Aβ42: two-tailed t-test, $n$(ctrl) = 10, $n$(FIRE) = 12, $p$ = 0.0665; insol. Aβ38: two-tailed t-test, $n$(ctrl) = 9, $n$(FIRE) = 12, $p$ = 0.017; insol. Aβ40: two-tailed t-test, $n$(ctrl) = 9, $n$(FIRE) = 12, $p$ = 0.0107; insol. Aβ42: two-tailed t-test with Welch's correction, $n$(ctrl) = 9, $n$(FIRE) = 12, $p$ = 0.00000007). **c** Representative images of amyloid plaques in the brains of 3-month-old control mice, FIRE mice, and FIRE mice xenografted with human microglia (FIRE + hMG), stained with X34 (fibrillar plaques) and 82E1 (total Aβ). **d** Quantifications of amyloid plaques in the whole brain (%X34 area: one-way Welch's ANOVA test, $n$(ctrl) = 11, $n$(FIRE) = 12, $n$(FIRE +hMG) = 7, $p$ = 0.000002; total number of X34 plaques: one-way Kruskal-Wallis test, $n$(ctrl) = 11, $n$(FIRE) = 12, $n$(FIRE +hMG) = 7, $p$ = 0.0001; average size X34 plaques: one-way ANOVA, $n$(ctrl) = 11, $n$(FIRE) = 12, $n$(FIRE +hMG) = 7, $p$ = 0.00000002; %82E1 area: one-way Welch's ANOVA test, $n$(ctrl) = 10, $n$(FIRE) = 12, $n$(FIRE +hMG) = 7, $p$ = 0.0000002; total number of 82E1 plaques: one-way Kruskal-Wallis test, $n$(ctrl) = 10, $n$(FIRE) = 12, $n$(FIRE +hMG) = 7, $p$ = 0.00002; average size 82E1 plaques: one-way Welch's ANOVA test, $n$(ctrl) = 10, $n$(FIRE) = 12, $n$(FIRE +hMG) = 7, $p$ = 0.000009). **e** Aβ ELISA of soluble and insoluble cortex extracts (sol. Aβ38: one-way ANOVA, $n$(ctrl) = 11, $n$(FIRE) = 12, $n$(FIRE +hMG) = 8, $p$ = 0.0237; sol. Aβ40: one-way ANOVA, $n$(ctrl) = 10, $n$(FIRE) = 12, $n$(FIRE +hMG) = 8, $p$ = 0.1868; sol. Aβ42: one-way ANOVA, $n$(ctrl) = 11, $n$(FIRE) = 12, $n$(FIRE +hMG) = 8, $p$ = 0.0125; insol. Aβ38: one-way ANOVA, $n$(ctrl) = 11, $n$(FIRE) = 12, $n$(FIRE +hMG) = 8, $p$ = 0.0000000001; insol. Aβ40: one-way ANOVA, $n$(ctrl) = 11, $n$(FIRE) = 11, $n$(FIRE +hMG) = 8, $p$ = 0.00000001; insol. Aβ42: one-way ANOVA, $n$(ctrl) = 11, $n$(FIRE) = 12, $n$(FIRE +hMG) = 8, $p$ < 0.000000000000001). White dots represent female mice and black dots represent male mice. Scale bars 500 μm (**a**, **c**) and 30 μm (**a**). All data are presented as mean ± SD. *$p$ ≤ 0.05; **$p$ ≤ 0.01; ***$p$ ≤ 0.001; ****$p$ ≤ 0.0001. Source data are provided as a Source Data file.

individually collected. The dissected brain regions were snap-frozen and stored at −80 °C until use for biochemistry. For sectioning, the PFA fixed hemisphere was embedded in 4% Top Vision Low Melting Point Agarose (Thermo Scientific) and sectioned coronally into 35–40 μm sections using a vibrating microtome (Leica). Brain sections were collected under free-floating conditions and stored in cryoprotectant solution (40% PBS, 30% ethylene glycol, 30% glycerol) at −20 °C.

For the APOE staining, sections were washed with 1x DPBS, and antigen retrieval was performed by incubation for 10 min in 10 mM sodium citrate at 95 °C, followed by 30 min incubation with RNAscope® Protease III (Advanced Cell Diagnostics, Cat.#322337) at 37 °C. All other stainings were performed without antigen retrieval and sections were immediately permeabilized for 30 min at room temperature in PBS with 0.2% Triton. After permeabilization, sections were stained with X-34 staining solution (10 μM X-34 (Sigma-Aldrich), 20 mM NaOH (Sigma-Aldrich), and 40% ethanol) for 20 min at room temperature. X34 is a derivative of Congo red and is a fluorescent beta-sheet specific amyloid dye. Sections were washed three times with 40% ethanol for 2 min and twice with PBS + 0.2% Triton for 5 min. Sections were blocked with 5% normal donkey serum in PBS + 0.2% Triton X-100 for 1 h at room temperature. Primary antibody incubation was done overnight at 4 °C. The 82E1 antibody reacts with the N-terminal of amyloid Aβ peptides and, therefore, also stains diffuse Aβ-aggregates in the brain. The next day, sections were washed 3x with PBS and incubated with secondary antibodies for 2 h at RT. Sections were washed and finally mounted with Glycergel mounting medium (Agilent). The details of all antibodies used for immunofluorescence are listed in Table 1. Confocal images were obtained using a Nikon AX microscope with at ×4 (NA 0.2), ×20 (NA 0,75), x40 (NA 1.25), and x60 (NA 1.42). All images were acquired using similar acquisition parameters such as 16-bit, 1024 × 1024 quality, and images were processed in the FIJI/Image J software. All the images of Z-series stacks were then converted to Fiji/ImageJ maximum intensity projections.

For quantification of amyloid plaques (X34 and 82E1) and dystrophic neurites (LAMP1) in the whole brain hemisphere, large images with roughly 4–6 z-stacks were obtained with a 4x (NA 0.2) objective. Microscope settings were kept the same for all samples that were compared. Z-stacks were then converted to Fiji/ImageJ maximum intensity projections. The brain hemisphere was manually outlined with the polygon selection tool and added to the ROI manager. An intensity threshold for amyloid plaque quantification was selected manually and used for all the sections. In some sections, the threshold had to be adjusted, e.g., due to increased background. In those cases, a second threshold was determined which was applied for all the sections showing increased background. After thresholding, particles were analyzed and the relative area, number, and average size of particles were recorded. Two sections per brain were analyzed and the average values per brain were calculated. The same untreated 7-month-old mice were used as the control for the respective 3 to 7 months and the 1 to 7 months PLX3397 treatment. A few samples were excluded due to damaged sections or imaging artifacts. Excluded data is indicated in the source data.

For the quantification of microglia depletion, three 20x images of different cortical areas were acquired per mouse. Z-stacks were converted to maximum intensity projections and IBA1 staining was automatically thresholded (Otsu dark). Particles were analyzed to determine IBA1$^+$ area.

## Isolation of soluble and insoluble brain extracts and Aβ ELISA

Soluble and insoluble brain extracts were isolated from snap-frozen cortex samples. Samples were randomized and 8 samples across different cohorts were processed together. Snap-frozen cortices were thawed on ice and tissue weights were recorded. Ten volumes (w/v) P-TER buffer (Thermo Fisher, Cat.#78510) supplemented with cOmplete™ Protease Inhibitor Cocktail (Roche, Cat.#5056489001) was added and the tissue was homogenized in Fast prep tubes (MP Biomedicals) for 45 s at 6.5 m/s. Samples were centrifuged for 5 min at 5000 × $g$ to remove debris. Subsequently, the supernatant was centrifuged for 1 h at 55,000 rpm at 4 °C in an Optima Ultracentrifuge using a TLA110 rotor (-125,000 × $g$) to pellet the insoluble brain fraction. The supernatant ( = soluble fraction) was collected and stored at -80 °C and the pellet was used for guanidine extraction.

Pellets were resuspended in 2 μl/mg tissue 6 M GuHCl solution (6 M GuHCl/50 mM Tris-HCl, protease inhibitor cocktail, pH 7.6). Samples were sonicated with a micro-tip for 30 s at 10% amplitude, vortexed for 5 min, and incubated on a shaker for 1 h at 25 °C and 450 rpm. Samples were ultracentrifuged for 20 min at 70,000 rpm and 4 °C (-230,000 × $g$). The supernatant, containing guanidine-soluble Aβ fractions (insoluble Aβ) were transferred into a new tube, diluted 12 times with GuHCl diluent (20 mM phosphate, 0.4 M NaCl, 2 mM EDTA, 10% Block Ace, 0.2% BSA, 0.05% NaN3, 0.075% CHAPS, protease inhibitor cocktail, pH 7.0), and stored at −80 °C until use.

Aβ38, Aβ40, and Aβ42 levels in the soluble and insoluble brain extracts were quantified by Meso Scale Discovery (MSD). Standard 96-well SECTOR plates (MSD, Cat.#L15XA-3) were coated with 1.5 μg/ml LTDA_38, LDA_40, or LTDA_Aβ42 capture antibody (homemade mouse monoclonal against Aβ$_{38}$, Aβ$_{40}$ or Aβ$_{42}$ neoepitope respectively) in PBS, pH 7.4 at 4 °C overnight. Plates were washed 5x with PBS-T (PBS + 0.05% Tween 20) and blocked with 0.1% casein in PBS for 1.5 h at room temperature. Aβ standard curves were prepared with human Aβ1-38 (rPeptide Cat.#A-1078-1), Aβ1-40 (Cat.#A-1153-1), or Aβ1-42 (Cat.#A-1163-1). Samples were diluted according to previously

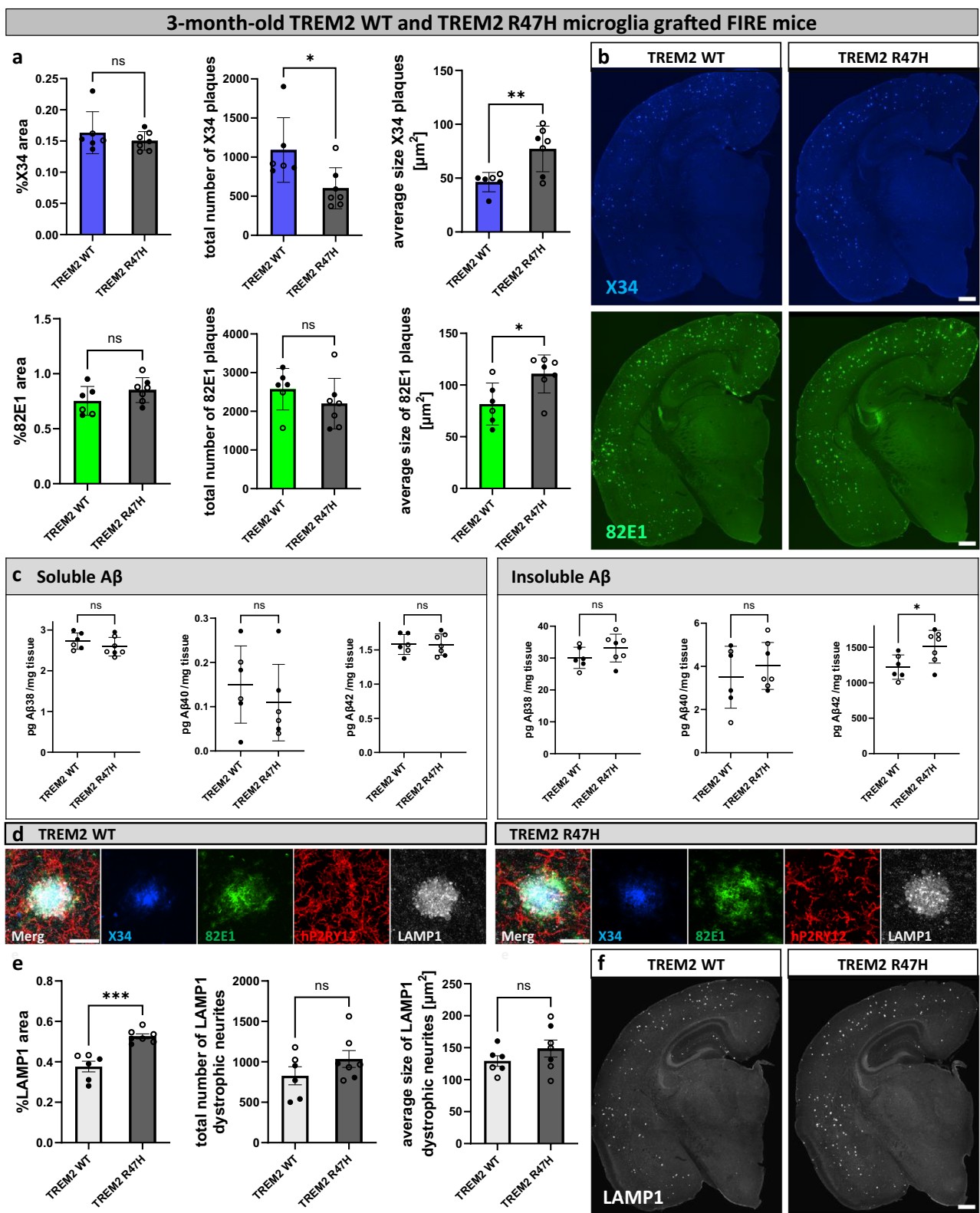

determined concentrations. Diluted samples and standards were mixed 1:1 with LTDA_hAβN labeled with a sulfo-TAG detection antibody (homemade mouse monoclonal against the N-terminal sequence of human Aβ, in collaboration with Maarten Dewilde), in 0.1% casein in PBS, loaded on the blocked MSD plate, and incubated overnight at 4 °C. Plates were washed 5x with PBS-T and 150 µl MSD GOLD Read Buffer A (Cat.#R92TG-2) was added to the wells. Plates were read with an MSD Sector Imager 2400 A. Outliers were identified in GraphPad

using ROUT (Q = 1%) and excluded from analysis. Outliers were likely caused by technical issues, e.g., air bubbles or bad coating of individual wells. All data is available in the source data including excluded measures and reasons for exclusion.

### Differentiation of microglial progenitors and transplantation

All experiments involving human stem cells have been approved by the UZ Leuven Ethical Committee and Biobank under study protocol

**Fig. 5 | Xenotransplanted *TREM2^{R47H/R47H}* microglia exacerbate amyloid and dystrophic neurite pathology in FIRE mice.** FIRE mice (*App^{NL-G-F}; Rag2^{-/-}; IL2rg^{-/-}; hCSF1^{KI}; Csf1r^{ΔFIRE/ΔFIRE}*) were xenotransplanted with human WT microglia (TREM2 WT) and human microglia harboring the *TREM2^{R47H/R47H}* risk gene (TREM2 R47H) at P4 and analyzed at 3 months of age. **a** Image quantification of X34⁺ and 82E1⁺ amyloid plaques in the whole brain (%X34 area: two-tailed Mann-Whitney test, n(TREM2 WT) = 6, n(TREM2 R47H) = 7, p = 0.6282; total number of X34 plaques: two-tailed Mann-Whitney test, n(TREM2 WT) = 6, n(TREM2 R47H) = 7, p = 0.014; average size X34 plaques: two-tailed t-test, n(TREM2 WT) = 6, n(TREM2 R47H) = 7, p = 0.0072; %82E1 area: two-tailed t-test, n(TREM2 WT) = 6, n(TREM2 R47H) = 7, p = 0.1734; total number of 82E1 plaques: two-tailed t-test, n(TREM2 WT) = 6, n(TREM2 R47H) = 7, p = 0.2909; average size 82E1 plaques: two-tailed t-test, n(TREM2 WT) = 6, n(TREM2 R47H) = 7, p = 0.0201). **b** Representative images of amyloid plaques in the brain stained with X34 (fibrillar plaques) and 82E1 (total Aβ). **c** ELISA of amyloid levels in soluble and insoluble cortex extracts (sol. Aβ38: two-tailed t-test, n(TREM2 WT) = 6, n(TREM2 R47H) = 7, p = 0.2712; sol. Aβ40: two-tailed t-test, n(TREM2 WT) = 6, n(TREM2 R47H) = 6, p = 0.434; sol. Aβ42: two-tailed t-test, n(TREM2 WT) = 6, n(TREM2 R47H) = 7, p = 0.9832; insol. Aβ38: two-tailed t-test, n(TREM2 WT) = 6, n(TREM2 R47H) = 7, p = 0.1897; insol. Aβ40: two-tailed t-test, n(TREM2 WT) = 6, n(TREM2 R47H) = 7, p = 0.4699; insol. Aβ42: two-tailed t-test, n(TREM2 WT) = 6, n(TREM2 R47H) = 7, p = 0.0295). **d** Zoom in on plaques, dystrophic neurites, and human microglia staining in 3-month-old xenografted FIRE mice. **e** Quantification of LAMP1⁺ dystrophic neurites in the whole brain (%LAMP1 area: two-tailed t-test, n(TREM2 WT) = 6, n(TREM2 R47H) = 7, p = 0.0002; total number of LAMP1 dystrophic neurites: two-tailed t-test, n(TREM2 WT) = 6, n(TREM2 R47H) = 7, p = 0.2023; average size of LAMP1 dystrophic neurites: two-tailed t-test, n(TREM2 WT) = 6, n(TREM2 R47H) = 7, p = 0.2574). **f** Representative images of LAMP1⁺ dystrophic neurites in the whole brain. White dots represent female mice and black dots represent male mice. Scale bars 500 μm (**b**, **f**), 30 μm (**d**). All data is presented as mean ± SD. *p ≤ 0.05; **p ≤ 0.01; ***p ≤ 0.001; ****p ≤ 0.0001. Source data are provided as a Source Data file.

S62888. H9-WT (WA09) human embryonic stem cells and H9 cells with *TREM2^{R47H/R47H}* mutation were differentiated into microglia progenitors and transplanted into the brains of *App^{NL-G-F}; Rag2^{-/-}; IL2rg^{-/-}; hCSF1^{KI}; Csf1r^{ΔFIRE/ΔFIRE}* mice following our published MIGRATE protocol[39]. H9 (WA09) cells were obtained from WiCell Research Institute and *TREM2^{R47H/R47H}* cells were acquired from the lab of Catherine M. Verfaillie at KU Leuven and have been generated from H9-WT (WA09) human embryonic stem cells as described by Claes et al[43].

In brief, control and *TREM2^{R47H/R47H}* H9-WA09 stem cells were plated and maintained on human Matrigel-coated six-well plates in E8 flex media until reaching ~70–80% confluence. Once confluent, stem cell colonies were dissociated into single cells using Accutase (Sigma-Aldrich) and plated into U-bottom 96-well plates at a density of ~10,000 per well in mTeSR1 medium with BMP4 (50 ng ml⁻¹), VEGF (50 ng ml⁻¹) and SCF (20 ng ml⁻¹) for 4 days and allowed them to self-aggregate into embryoid bodies. On day 4, embryoid bodies were transferred into six-well plates (~20 embryoid bodies per well) in X-VIVO (LO BE02-060F, Westburg) (+ supplements) medium supplemented with SCF (50 ng ml⁻¹), M-CSF (50 ng ml⁻¹), IL-3 (50 ng ml⁻¹), FLT3 (50 ng ml⁻¹) and TPO (5 ng ml⁻¹) for 7 days. A full

medium change was performed on day 8. On day 11, the differentiation medium was replaced with X-VIVO (+ supplements) with FLT3 (50 ng ml⁻¹), M-CSF (50 ng ml⁻¹) and GM-CSF (25 ng ml⁻¹). On day 18, floating human microglial precursors were collected from the supernatant and engrafted into P4 mouse brains (0.5 million cells per pup) by bilateral injection with a Hamilton syringe as previously described[39]. Heterozygous *App^{NL-G-F}; Rag2^{-/-}; IL2rg^{-/-}; hCSF1^{KI}; Csf1r^{+/ΔFIRE}* females and homozygous *App^{NL-G-F}; Rag2^{-/-}; IL2rg^{-/-}; hCSF1^{KI}; Csf1r^{ΔFIRE/ΔFIRE}* males were used for breeding, or pups were fostered with CD1 mothers when breeding with *App^{NL-G-F}; Rag2^{-/-}; IL2rg^{-/-}; hCSF1^{KI}; Csf1r^{ΔFIRE/ΔFIRE}* homozygous females to facilitate the survival.

### Statistics and reproducibility

Statistical tests and data visualization were performed using GraphPad Prism10. Each data point represents one mouse and data is presented as mean ± SD. Animals were randomly assigned to conditions to account for potential ordering effects. For all the ELISA data, statistical outliers (caused by technical errors) were identified using the ROUT test in Prism10 (Q = 1%) and excluded from further analysis. 2–3 brain sections per mouse were analyzed and averaged for histological analysis. To avoid litter bias in the mouse experiments, experimental groups were composed of animals from different litters randomly distributed. Analysis was performed semi-automated or fully automated (when possible) using FIJI/Image J. Normality of residuals was checked with the Shapiro−Wilk tests and homoscedasticity was checked with the F-test or Brown-Forsythe test (for three groups). Comparisons between two groups following a normal distribution were analyzed using a two-tailed unpaired t-test, when homoscedasticity requirements were not met Welch's correction was applied. Comparisons between two groups not following a normal distribution were analyzed with the Mann-Whitney test. When three groups were compared and data were normally distributed and homoscedastic, ordinary one-way ANOVA was used; when significant, it was followed by Tukey's multiple comparisons test. When data were normally distributed but not homoscedastic, Welch's ANOVA was used; when significant, it was followed by Dunnett's multiple comparison. When data were not normally distributed, ranks were compared with Kruskal-Wallis followed by Dunn's multiple comparisons test. Statistical significance was set at p < 0.05. All data necessary for the conclusions of the study are available in the main text, figures, and Supplementary Figs. Source data, including excluded data and reasons for exclusion, and detailed statistical analysis are provided with this paper.

### Reporting summary

Further information on research design is available in the Nature Portfolio Reporting Summary linked to this article.

### Table 1 | Antibodies used for immunofluorescence stainings

| Antibody | Source | Identifier | dilution |
| --- | --- | --- | --- |
| Mouse anti-82E1 | IBL | 10323 | 1/200 |
| Rabbit anti-Iba1 | WAKO | 019-19741 | 1/200 |
| Guinea pig anti-Iba1 | Synaptic Systems | 234 308 | 1/500 |
| Rat anti-Lamp1 | Santa Cruz | sc-19992 | 1/500 |
| Rabbit anti-ubiquitin | Abcam | ab134953 | 1/250 |
| Mouse anti-human CD9 | BioLegend | 312102 | 1/500 |
| Rabbit anti-human P2RY12 | Atlas antibodies | HPA014518 | 1/500 |
| Goat anti-APOE | Chemicon International | AB947 | 1/1000 |
| Donkey anti-mouse Alexa Fluor 488 | Invitrogen | A21202 | 1/500 |
| Donkey anti-rabbit Alexa Fluor 594 | Invitrogen | A21207 | 1/500 |
| Donkey anti-rat Alexa Fluor 647 | Abcam | ab150155 | 1/500 |
| Donkey anti-rabbit Alexa Fluor 488 | Invitrogen | A21206 | 1/500 |
| Donkey anti-mouse Alexa Fluor 594 | Invitrogen | A21203 | 1/500 |
| Donkey anti-guinea pig Cy5 | Jackson Immunolabs | 706-175-148 | 1/500 |
| Donkey anti-goat Alexa Fluor 647 | Invitrogen | A21447 | 1/500 |

## Data availability

All data necessary for the conclusions of the study are available in the main text, figures and Supplementary Figures. Source data are provided as a Source Data file. Source data are provided with this paper.

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

## Acknowledgements

The authors thank Prof. Catherine Verfaillie for the *TREM2^{R47H/R47H}* cells, Prof. Maarten Dewilde for the antibodies used for Aβ ELISA's. An Snellinx for help with the transplantation work. Amber Claes and Veronique Hendricks for breeding and taking care of the mice and the VIB Mouse Expertize Unit for generation of the FIRE mice. This work was funded by a Medical Research Grant (MR/Y014847/1) awarded to B.D.S., the European Research Council (ERC) under the European Union's Horizon 2020 Research and Innovation Programme (grant agreement no. ERC-834682 CELLPHASE_AD). The Flanders Institute for Biotechnology (VIB vzw), a Methusalem grant from KU Leuven and the Flemish Government, the Fonds voor Wetenschappelijk Onderzoek, KU Leuven, The Queen Elisabeth Medical Foundation for Neurosciences, the Opening the Future campaign of the Leuven Universitair Fonds, The Belgian Alzheimer Research Foundation (SAO-FRA) and the Alzheimer's Association USA. B.D.S. holds the Bax-Vanluffelen Chair for Alzheimer's Disease. N.B. is the recipient of a PhD fellowship from Fonds voor Wetenschappelijk Onderzoek (fellowship no. 11PSA24N). Schemes in Figs. 1a and 2a, and Supplementary Figs. 1d and 2c were created in BioRender.

## Author contributions

N.B., S.B., and B.D.S conceived and designed the study and wrote the manuscript. N.B. performed experiments and analyzed data. G.A. and S.C.B. performed experiments involving microglia transplantation. L.S. generated the FIRE mouse line with the assistance of CP. All authors discussed the results and commented on the manuscript.

## Competing interests

B.D.S. has been a consultant for Eli Lilly, Biogen, Janssen Pharmaceutica, Eisai, AbbVie and other companies and is now consultant to Muna Therapeutics. B.D.S is a scientific founder of Augustine Therapeutics and a scientific founder and stockholder of Muna Therapeutics. The remaining authors declare no competing interests.
