## [Transparent Peer Review file · Nature Communications]

Homeostatic microglia initially seed and activated microglia later reshape amyloid plaques in Alzheimer's Disease

Corresponding Author: Professor Bart De Strooper

Version 0:

Reviewer comments:

Reviewer #1

(Remarks to the Author)

Baligács and colleagues used the CSF1R inhibitor PLX3397 to deplete resident microglia, and likely perivascular macrophages, in the AppNL-G-F model from 1 to 4 months of age. Through immunofluorescence and ELISA, they observed a decrease in insoluble amyloid plaques, corresponding with fewer amyloid-related dystrophic neurites. To further investigate, they applied a microglia-deficient model (FIRE mice, *Csf1r* Δ FIRE/ Δ FIRE) alongside human microglia xenotransplantation, highlighting microglia's role in amyloid plaque development, as xenotransplantation reintroduced amyloid deposition. Additionally, they compared TREM2 WT and TREM2 R47H microglia's effects on amyloid formation in the AppNL-G-F; *Rag2*^{-/-}; *IL2rg*^{-/-}; *hCSF1KI*; *Csf1r* Δ FIRE/ Δ FIRE mouse model. Results, including minimal impact from late-stage PLX3397 treatment, suggest microglia aid in early amyloid seeding and modify preformed plaques later on. Supplementary figures provide further pathological and human microglia transplantation characterizations. Although the concept that microglia promote early plaque formation has been established in previous studies, the study is well executed and data are convincing.

1. the authors should include some mechanistic experiments to explain why TREM2 R47H+ microglia are less effective in plaque seeding. Defects in survival, phagocytosis, lysosomal activity, metabolism?
2. the authors should discuss their results in the context of previous studies, highlighting what is consistent and what is different

Reviewer #2

(Remarks to the Author)

Review of Baligacs et al

Response to previous reviews:

Monocyte derived microglia entering the CNS would be detected by *Iba1*; the counterargument that they are not observed seems compelling.

The argument that they do not look at individual cells because they depend on digital imaging is not convincing: as Bob Terry once observed, "automated imaging automatically makes mistakes". It would be trivial to confirm their observations with some manual counts, and they should do so.

The argument that apoE is not involved because the immunostaining does not detect apoE in microglia is also weak. Immunostaining rarely detects the secreted apoE molecules in their cells of origin. Homeostatic microglia also make apoE, and whether a microglia in the immediate vicinity of a nascent plaque (as suggested by the current work) secretes apoE into the immediate vicinity seems quite plausible. The cited work in astrocytes knockout does not, obviously, perform the experiment as Baligacs has done, and so the carry over of their conclusions is not a strong conclusion. The discussion should be more circumspect. Given the extraordinary effect of apoE null on plaque generation, and the unexpected but extremely strong difference between human and murine apoE on plaque generation, it still seems as if a role for local apoE in plaque initiation has not been ruled out as a mechanism. Indeed, speculation about molecular mechanism of the observed

phenomenon is largely missing, and the authors are encouraged to extend the discussion a bit.

Overall this manuscript is improved from the original, and does contain interesting data that suggest in elegant experiments that microglia do have multiple roles at different points in the disease. The discussion would be improved by both a more detailed exploration of mechanism, and by the simple observation that in AD brain, both the “initiation” and “mature” aspect of plaque formation are likely occurring simultaneously, complicating therapeutic approaches to either stimulate or block microglial activation.

Point-by-point response to reviewer reports

REVIEWERS' COMMENTS

Reviewer #1 (Remarks to the Author):

Baligács and colleagues used the CSF1R inhibitor PLX3397 to deplete resident microglia, and likely perivascular macrophages, in the AppNL-G-F model from 1 to 4 months of age. Through immunofluorescence and ELISA, they observed a decrease in insoluble amyloid plaques, corresponding with fewer amyloid-related dystrophic neurites. To further investigate, they applied a microglia-deficient model (FIRE mice, *Csf1r* Δ FIRE/ Δ FIRE) alongside human microglia xenotransplantation, highlighting microglia's role in amyloid plaque development, as xenotransplantation reintroduced amyloid deposition. Additionally, they compared TREM2 WT and TREM2 R47H microglia's effects on amyloid formation in the AppNL-G-F; *Rag2*^{-/-}; *IL2rg*^{-/-}; *hCSF1KI*; *Csf1r* Δ FIRE/ Δ FIRE mouse model. Results, including minimal impact from late-stage PLX3397 treatment, suggest microglia aid in early amyloid seeding and modify preformed plaques later on. Supplementary figures provide further pathological and human microglia transplantation characterizations.

Although the concept that microglia promote early plaque formation has been established in previous studies, the study is well executed and data are convincing.

1. the authors should include some mechanistic experiments to explain why TREM2 R47H+ microglia are less effective in plaque seeding. Defects in survival, phagocytosis, lysosomal activity, metabolism?
2. the authors should discuss their results in the context of previous studies, highlighting what is consistent and what is different

Authors response:

We thank the referee for summarizing our work and their valuable feedback.

1. We agree that mechanistic experiments on how microglia are seeding plaques and how the seeding may be affected by different AD risk genes would be very valuable. We intend to follow up with further studies. On the other hand, we believe that the current work is a complete study that settles an important discussion in the field.

We hypothesize that plaque seeding is mediated by uptake and aggregation of A β in the endo-lysosomal system of microglia. Many genetic risk factors for AD also impact endocytosis, phagocytosis, and the lysosomal system and may increase the seeding capacity of microglia. We slightly expanded the discussion in our manuscript to clarify.

2. We summarized previous studies and indicated which of the findings are consistent with our results in the supplementary Table. In response to the request of reviewer 1, we refer to this table more extensively in the new version of the paper.

Reviewer #2 (Remarks to the Author):

Review of Baligacs et al

Response to previous reviews:

Monocyte derived microglia entering the CNS would be detected by IBA1; the counterargument that they are not observed seems compelling.

The argument that they do not look at individual cells because they depend on digital imaging is not convincing: as Bob Terry once observed, “automated imaging automatically make mistakes”. It would be trivial to confirm their observations with some manual counts, and they should do so.

The argument that ApoE is not involved because the immunostaining does not detect ApoE in microglia is also weak. Immunostaining rarely detects the secreted ApoE molecules in their cells of origin. Homeostatic microglia also make ApoE, and whether a microglia in the immediate vicinity of a nascent plaque (as suggested by the current work) secretes ApoE into the immediate vicinity seems quite plausible. The cited work in astrocytes knockout does not, obviously, perform the experiment as Baligacs has done, and so the carry over of their conclusions is not a strong conclusion. The discussion should be more circumspect. Given the extraordinary effect of ApoE null on plaque generation, and the unexpected but extremely strong difference between human and murine ApoE on plaque generation, it still seems as if a role for local ApoE in plaque initiation has not been ruled out as a mechanism. Indeed, speculation about molecular mechanism of the observed phenomenon is largely missing, and the authors are encouraged to extend the discussion a bit.

Overall this manuscript is improved from the original, and does contain interesting data that suggest in elegant experiments that microglia do have multiple roles at different points in the disease. The discussion would be improved by both a more detailed exploration of mechanism, and by the simple observation that in AD brain, both the “initiation” and “mature” aspect of plaque formation are likely occurring simultaneously, complicating therapeutic approaches to either stimulate or block microglial activation.

Authors response:

We agree with the reviewer that counting individual microglia would give a more accurate estimate of microglia depletion efficiency. Our automated analysis of IBA1 coverage may be affected by microglia cell states and morphologies. However, quantification of the relative area covered by IBA1 staining is an established method to validate microglia depletion in the brain¹⁻³. Further, a more accurate estimate of microglia depletion efficiency would not affect the conclusions of our study.

Regarding the contribution of ApoE to amyloid plaque formation in microglia, we agree that ApoE is important in amyloid plaque seeding by microglia as recent work confirmed⁴. However, we hypothesize that ApoE is secreted by astrocytes, the primary providers of ApoE in the CNS⁵, and then taken up by microglia into the endo-lysosomal system where it serves as a co-factor for plaque formation⁴. This is supported by the fact that microglia-specific ApoE deletion does not alter plaque formation in mice⁶. We tried to clarify this by expanding the discussion of our manuscript.

Lastly, we also agree that the seeding and compaction of plaques by microglia may occur simultaneously. However, we suggest that stimulation of microglia may be beneficial for both aspects. Future work is needed to confirm and explore the molecular mechanisms.

References

1. Delizannis, A. T. *et al.* Effects of microglial depletion and TREM2 deficiency on A β plaque burden and neuritic plaque tau pathology in 5XFAD mice. *Acta Neuropathol Commun* **9**, (2021).
2. Sosna, J. *et al.* Early long-term administration of the CSF1R inhibitor PLX3397 ablates microglia and reduces accumulation of intraneuronal amyloid, neuritic plaque deposition and pre-fibrillar oligomers in 5XFAD mouse model of Alzheimer's disease. *Mol Neurodegener* **13**, 1–11 (2018).
3. Clayton, K. *et al.* Plaque associated microglia hyper-secrete extracellular vesicles and accelerate tau propagation in a humanized APP mouse model. *Mol Neurodegener* **16**, (2021).
4. Kaji, S. *et al.* Apolipoprotein E aggregation in microglia initiates Alzheimer's disease pathology by seeding β -amyloidosis. *Immunity* (2024) doi:10.1016/j.immuni.2024.09.014.
5. Xu, Q. *et al.* Profile and regulation of apolipoprotein E (ApoE) expression in the CNS in mice with targeting of green fluorescent protein gene to the ApoE locus. *Journal of Neuroscience* **26**, 4985–4994 (2006).
6. Henningfield, C. M., Arreola, M. A., Soni, N., Spangenberg, E. E. & Green, K. N. Microglia-specific ApoE knock-out does not alter Alzheimer's disease plaque pathogenesis or gene expression. *Glia* **70**, 287–302 (2022).